# Nuclear export of misfolded SOD1 mediated by a normally buried NES-like sequence reduces proteotoxicity in the nucleus

Yongwang Zhong[1,2], Jiou Wang[3,4], Mark J Henderson[5], Peixin Yang[6,7], Brian M Hagen[1,2], Teepu Siddique[8], Bruce E Vogel[1,2], Han-Xiang Deng[8], Shengyun Fang[1,2,6*]

[1]Center for Biomedical Engineering and Technology, University of Maryland School of Medicine, Baltimore, United States; [2]Department of Physiology, University of Maryland School of Medicine, Baltimore, United States; [3]Department of Biochemistry and Molecular Biology, Johns Hopkins University, Baltimore, United States; [4]Department of Neuroscience, Johns Hopkins University, Baltimore, United States; [5]National Center for Advancing Translational Sciences, National Institutes of Health, Rockville, United States; [6]Department of Biochemistry and Molecular Biology, University of Maryland School of Medicine, Baltimore, United States; [7]Department of Obstetrics, Gynecology and Reproductive Sciences, University of Maryland School of Medicine, Baltimore, United States; [8]Division of Neuromuscular Medicine, Davee Department of Neurology and Clinical Neurosciences, Northwestern University Feinberg School of Medicine, Chicago, United States

*For correspondence: sfang@umaryland.edu

Competing interests: The authors declare that no competing interests exist.

**Abstract** Over 170 different mutations in the gene encoding SOD1 all cause amyotrophic lateral sclerosis (ALS). Available studies have been primarily focused on the mechanisms underlying mutant SOD1 cytotoxicity. How cells defend against the cytotoxicity remains largely unknown. Here, we show that misfolding of ALS-linked SOD1 mutants and wild-type (wt) SOD1 exposes a normally buried nuclear export signal (NES)-like sequence. The nuclear export carrier protein CRM1 recognizes this NES-like sequence and exports misfolded SOD1 to the cytoplasm. Antibodies against the NES-like sequence recognize misfolded SOD1, but not native wt SOD1 both in vitro and in vivo. Disruption of the NES consensus sequence relocalizes mutant SOD1 to the nucleus, resulting in higher toxicity in cells, and severer impairments in locomotion, egg-laying, and survival in *Caenorhabditis elegans*. Our data suggest that SOD1 mutants are removed from the nucleus by CRM1 as a defense mechanism against proteotoxicity of misfolded SOD1 in the nucleus.

## Introduction

Amyotrophic lateral sclerosis (ALS) is a fatal neurodegenerative disease caused by progressive loss of upper and lower motor neurons in the brain and spinal cord. Most patients die from respiratory failure in 3–5 years after disease onset (*Ajroud-Driss and Siddique, 2015*). About 10% of ALS cases have positive family history and are classified as familial ALS (fALS). The remaining 90% of ALS cases are classified as sporadic ALS (sALS). Mutations in over 40 genes have been associated with fALS (*Ajroud-Driss and Siddique, 2015*), including the first identified gene encoding Cu/Zn superoxide dismutase (SOD1) (*Deng et al., 1993*; *Rosen et al., 1993*). To date, over 170 mutations involving 88

**eLife digest** Amyotrophic lateral sclerosis (ALS) is a disease that leads to muscle weakness and paralysis. The symptoms become progressively worse over time to the point that patients die because they become unable to breathe. Over 170 different genetic mistakes (or mutations) in a gene that encodes a protein called SOD1 are known to cause ALS. These mutations cause the SOD1 protein to form different shapes that are toxic to nerve cells, leading to the gradual loss of the nerve cells that control movement. SOD1 is normally found in a compartment within nerve cells called the nucleus, which is where most of the cell's genetic information is stored and managed.

A nematode worm called *Caenorhabditis elegans* has often been used as a model to study the role of SOD1 in ALS because its nervous system shares many features in common with ours but is much smaller. Some evidence suggests that cells may be able to defend themselves against the harmful effects of abnormal SOD1 proteins. However, it is not clear how these defences might work.

Zhong et al. examined variants of SOD1 proteins from human cells grown in a laboratory. The experiments show that some mutant SOD1 proteins fold in such a way that a small section of the protein that is normally buried within the protein's structure is exposed on the surface. Mutant SOD1 proteins that expose this "peptide" are removed from the nucleus and are linked with faster progression of ALS in patients.

Further experiments show that another protein called CRM1 can recognise this exposed peptide, leading to the removal of the mutant SOD1 proteins from the nucleus. Zhong et al. found that if mutant SOD1 is not removed from the nucleus of nerve cells, the nematode worms developed ALS symptoms even faster.

These findings suggest that cells may be able to remove some mutant SOD1 proteins from the nucleus to defend themselves against the proteins' toxic effects. Future work will reveal whether other cells use this approach to protect themselves against other diseases. The peptide discovered in this work may also have the potential to be used as a marker to predict how individual cases of ALS will progress, or as a target for treatments against the disease.

of the 154 amino acids of SOD1 have been identified in patients diagnosed with ALS (*Taylor et al., 2016*) (alsod.iop.kcl.uk). Mutations in SOD1 account for about 20% of familial and 2–7% of sporadic ALS cases (alsod.iop.kcl.uk), which makes SOD1 the second most frequently mutated gene after C9ORF72 in affected Caucasians (*Ajroud-Driss and Siddique, 2015*; *Renton et al., 2014*).

SOD1 is a highly conserved and ubiquitously expressed free-radical scavenging enzyme. It catalyzes the dismutation of superoxide radical ($O_2 \cdot^-$) into oxygen and hydrogen peroxide ($H_2O_2$), which in turn is reduced to water and oxygen by catalase. SOD1 is highly abundant, comprising 1–2% of the total soluble protein in the nervous system (*Pardo et al., 1995*). Despite its important physiological function, SOD1 knockout mice failed to develop an ALS-like phenotype (*Reaume et al., 1996*). Increasing evidence indicates that a toxic gain-of-function in SOD1 mutants is a fundamental mechanism for the pathogenesis of ALS (*Williamson et al., 2000*; *Deng et al., 2006*; *Sau et al., 2007*; *Rotunno and Bosco, 2013*; *Bunton-Stasyshyn et al., 2015*; *Silverman et al., 2016*). Although the exact mechanisms by which diverse SOD1 mutations all cause ALS remains unclear, many toxic effects caused by SOD1 mutants have been reported, including excitotoxicity, oxidative stress, endoplasmic reticulum (ER) stress, mitochondrial dysfunction, axonal transport disruption, prion-like propagation, and non-cell autonomous toxicity of neuroglia (*Hayashi et al., 2016*). These toxic effects may contribute to mutant SOD1-induced neuronal degeneration. Aggregation of SOD1 mutants in motor neurons is a common pathological feature. Therefore, coaggregation of an unidentified essential component or components or aberrant catalysis by misfolded mutants may underlie part of the mutant-mediated toxicity (*Bruijn et al., 1998*).

Disease pathogenesis and progression is not only determined by the toxic effects of SOD1 mutants but also modulated by intrinsic cellular defenses against the toxic proteins, such as protein quality control. Indeed, it has been reported that removal of SOD1 mutants through degradation by the ubiquitin-proteasome system and autophagy, or through induction of heat shock proteins reduces their cytotoxicity (*Kabuta et al., 2006*; *Tan et al., 2008*; *Ying et al., 2009*; *Crippa et al.,*

*2010*; *Kalmar et al., 2014*). However, available studies have been primarily focused on how mutant SOD1 exerts its toxicity, and much less is known about how cells defend against the toxic effects. In this study, we identified a novel mechanism that reduces mutant SOD1 cytotoxicity through limiting proteotoxicity in the nucleus.

## Results

### ALS-linked SOD1 mutants are cleared from the nucleus by CRM1

SOD1 is normally localized in both the cytoplasm and the nucleus of cells (*Crapo et al., 1992*). We found that when GFP-tagged wt-SOD1 (GFP-SOD1$^{wt}$) and two mutants (GFP-SOD1$^{G85R}$ and GFP-SOD1$^{G93A}$) were expressed in HeLa cells, both mutants showed predominantly cytoplasmic distribution while GFP-SOD1$^{wt}$ was evenly distributed in the cytoplasm and the nucleus as expected (*Figure 1A*). Impairment of the CRM1-dependent nuclear export pathway with a CRM1 inhibitor, leptomycin B (LMB), caused the mutants to accumulate in the nucleus (*Figure 1A*). Similarly, knockdown of CRM1 expression by RNAi or co-expression of a Q69L mutant of Ran GTPase, a dominant-negative inhibitor of CRM1 (*Xu et al., 2010*; *Kehlenbach et al., 1999*), also led to accumulation of the mutants in the nucleus (*Figure 1B,C*). To rule out the contribution of proteasomal degradation to the nuclear clearance of SOD1 mutants (*Kabuta et al., 2006*; *Niwa et al., 2002*), we inhibited the proteasome activity with a proteasome inhibitor MG132 along with inhibition of protein synthesis by cycloheximide (CHX). Proteasome inhibition for 60 min did not significantly affect the nuclear levels of GFP-SOD1$^{G85R}$ but again LMB treatment did (*Figure 1D*). These results suggest that the lack of nuclear distribution of SOD1 mutants is mediated by CRM1-dependent nuclear export.

### A nuclear export signal (NES)-like peptide sequence is responsible for the nuclear clearance of SOD1 mutants

It is well known that CRM1 exports its physiological cargo proteins through the recognition of the leucine-rich nuclear export signal (NES) (*Kosugi et al., 2008*; *Xu et al., 2012*). Selective export of mutant but not wt SOD1 suggests that SOD1 does not have a physiological NES. We hypothesized that misfolding of SOD1 mutants reveals a normally buried hydrophobic peptide that contains a NES consensus sequence. To test this hypothesis, we analyzed SOD1 protein sequence and located two potential NES-like sequence-containing peptides (*Figure 2A*). The peptides were each fused to mCherry and expressed in HeLa cells. Fusion with peptide P1 corresponding to residues 24–55 of SOD1, but not with P2 resulted in the nuclear clearance of mCherry, which could be restored by LMB treatment (*Figure 2B*). GST pull-down experiments showed that P1 and the NES from protein kinase inhibitor (PKI) (*Güttler et al., 2010*) as a positive control, but not P2, interact with CRM1 protein (*Figure 2C*). These results suggest that peptide P1 in human SOD1 contains a functional NES consensus sequence.

Next, we asked if the NES activity of P1 sequence is responsible for the nuclear export of SOD1 mutants. Eight hydrophobic residues in P1 are among the candidates for the NES consensus residues, and therefore, each was mutated to Arginine (R) in GFP-SOD1$^{G93A}$. Five of the resulting mutants, including I35R, L38R, L42R, F45R, and V47R, showed even distribution in the cytoplasm and the nucleus, suggesting these five hydrophobic residues are essential for G93A nuclear export (*Figure 2D*). These essential residues are in good agreement with the PKI-class NES consensus $\Phi^0$-$X_2$-$\Phi^1$-$X_3$-$\Phi^2$-$X_{2-3}$-$\Phi^3$-$X$-$\Phi^4$ (*Güttler et al., 2010*) (*Figure 2E*), where $\Phi^0$-$\Phi^4$ are five key hydrophobic residues spaced by different number of amino acids (denoted as $X_n$) that are preferentially charged, polar or small. Importantly, sequence alignment revealed that one of the five essential hydrophobic residues, Leu42 ($\Phi^2$), is not conserved in orthologs for human SOD1 (*Figure 2E*). Mutation of this residue to Glutamine (Q), as is naturally present in the mouse ortholog, abolished the nuclear export ability of GFP-SOD1$^{G93A}$ (*Figure 2F*). These results indicate that P1 is not an evolutionarily conserved physiological NES but may be exposed by misfolding of SOD1 mutants.

### The NES-like sequence is normally buried but exposed in SOD1 mutants

To determine whether the NES-like sequence is normally buried, we analyzed the available structure of wt-SOD1 (PDB: 2c9v). The NES-like consensus sequence (aa35-47) forms part of strand $\beta$3, the crossover loop, and strand $\beta$4, and is almost completely buried in the structure (*Figure 3A*). In fact,

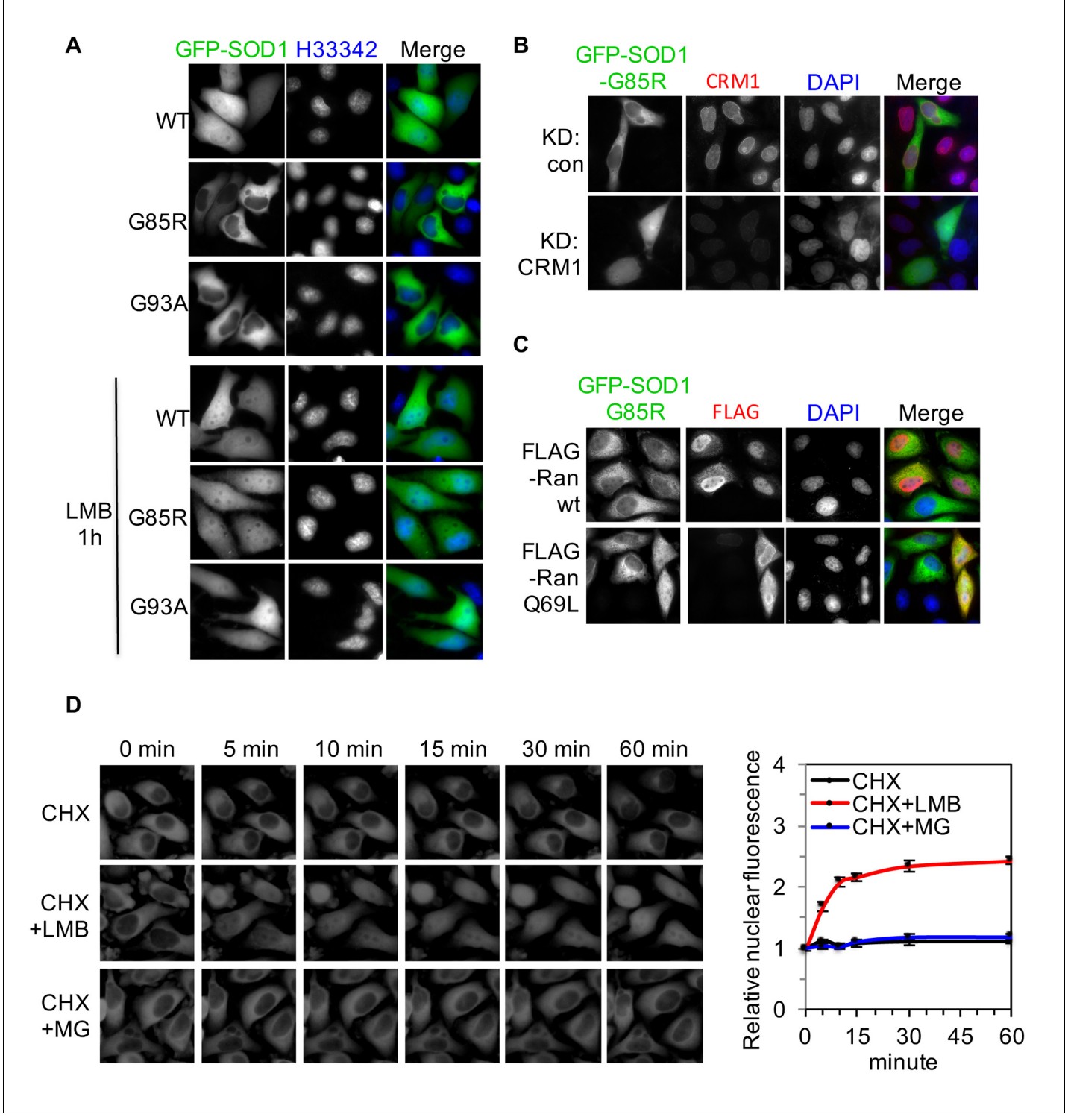

**Figure 1.** ALS-linked SOD1 mutants are exported from the nucleus by CRM1. (**A**) Inhibition of CRM1-dependent nuclear export increases nuclear distribution of SOD1$^{G85R}$ and SOD1$^{G93A}$. HeLa cells expressing GFP-tagged SOD1$^{wt}$, SOD1$^{G85R}$ or SOD1$^{G93A}$ were treated with LMB (20 nM) for 1 hr. (**B**) Knockdown of CRM1 increases nuclear distribution of GFP-SOD1$^{G85R}$. (**C**) Overexpression of FLAG-tagged RanQ69L but not wt Ran increases nuclear distribution of GFP-SOD1$^{G85R}$. (**D**) Time-lapse imaging of SOD1$^{G85R}$. HeLa cells expressing GFP-SOD1$^{G85R}$ were treated with cycloheximide (CHX, 100 nM), in combination with MG132 (30 μM) or LMB (20 nM) as indicated. Images were acquired at indicated time points. The nuclear (N) and total (T) GFP fluorescence was measured for each cell with ImageJ. The N/T ratio expressed as mean ± SEM was calculated from 30 cells for each group and plotted. The mean N/T ratios at start point (0 min) were arbitrary set as 1. Paired $t$-test, CHX vs CHX+LMB: p=0.008; CHX+LMB vs CHX+MG132: p=0.006; CHX vs CHX+MG132: p=0.494.

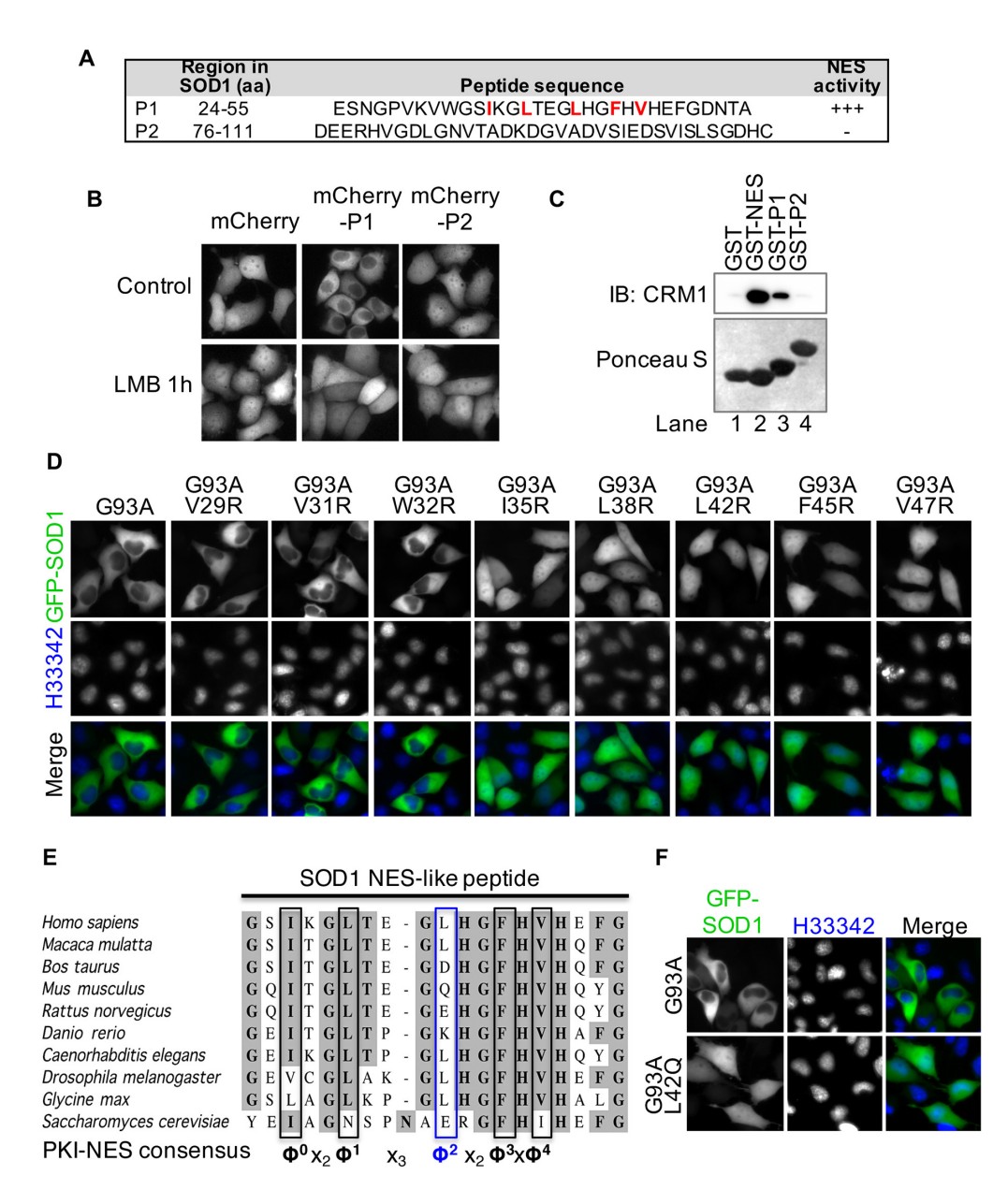

**Figure 2.** A nuclear export signal (NES)-like sequence is essential for the nuclear export of SOD1 mutants. (**A**) Two putative regions containing NES consensus sequences in SOD1 (P1 and P2). (**B**) P1 or P2 fused with mCherry was expressed in HeLa cells and treated with LMB for 1 hr. (**C**) GST pull-down. Immobilized GST or GST-tagged PKI-NES (NES), P1 or P2 was incubated with recombinant CRM1 and RanGTP. (**D**) Identification of the key hydrophobic residues required for nuclear export activity in P1. Each of eight hydrophobic residues in P1 was mutated to Arginine in GFP-SOD1[G93A]. The mutants were expressed in HeLa cells. (**E**) Alignment of key residues in P1 with corresponding residues in human SOD1 orthologs and PKI-class NES consensus (*Güttler et al., 2010*). Note that residue Leu42 in human ($\Phi^2$) is not conserved in several other species, including mouse. (**F**) Mutation of Leu42 in GFP-tagged human SOD1[G93A] to mouse corresponding residue (Gln) abolishes nuclear export of the mutant SOD1.

Leu42 is the only one of the five essential hydrophobic residues exposed on the protein surface (*Figure 3A*), which may explain why natural substitution of Leu42 by Gln in mouse SOD1 does not affect its structure (*Figure 2E*). Next, we assessed the accessibility of the NES-like sequence to CRM1 in SOD1 mutants using pull-down assays with recombinant GST-tagged full-length wt or mutant SOD1. CRM1 was precipitated by all four ALS-linked mutants tested, including Q22L, G85R,

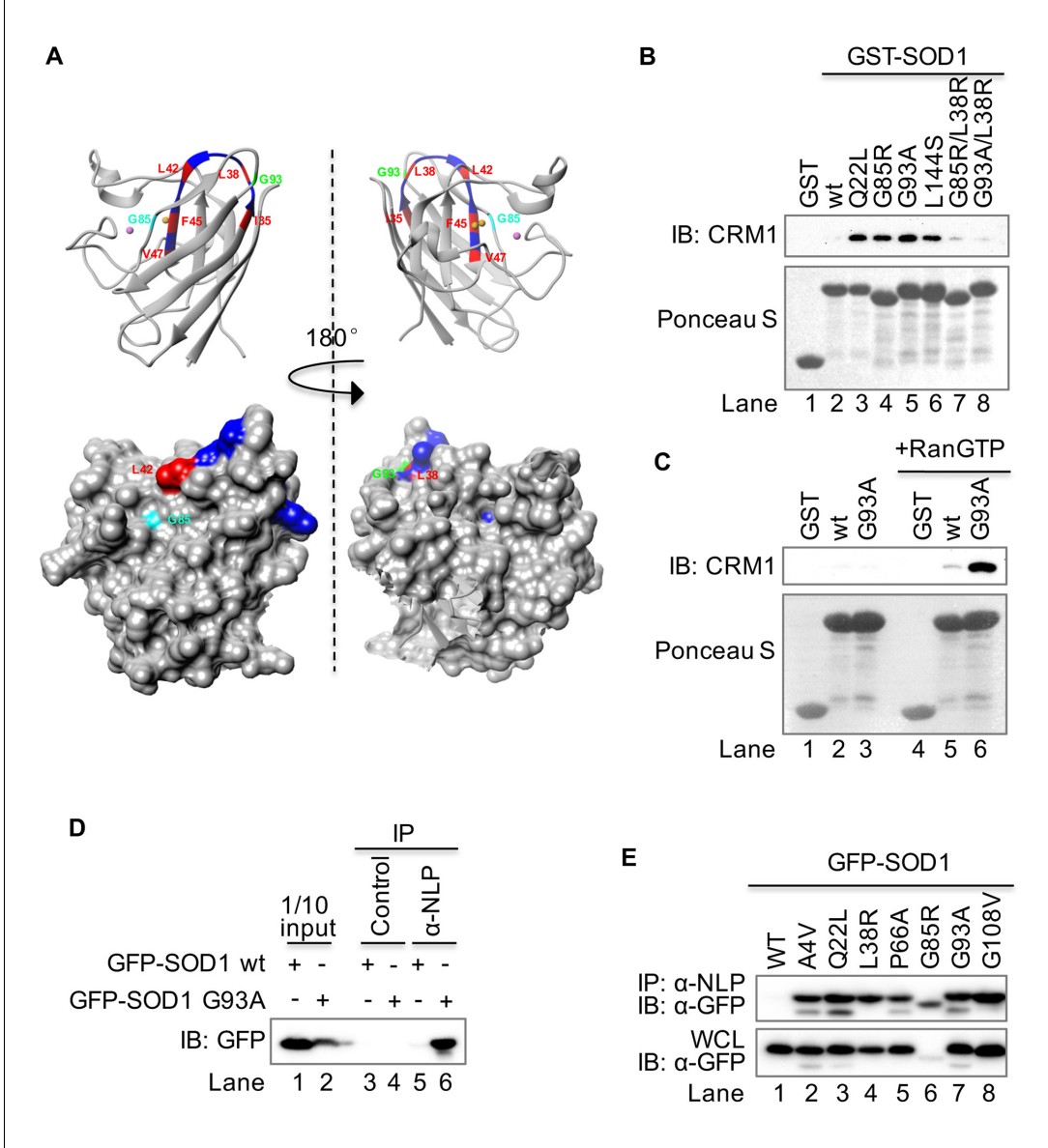

**Figure 3.** The NES-like sequence is exposed only in SOD1 mutants but not in wt SOD1. (**A**) Structural localization of the NES-like peptide in wt SOD1. The X-ray crystallographic structure of wt SOD1 (PDB: 2c9v, only chain A of the dimer was shown) is analyzed in UCSF Chimera 1.8. Key residues in the NES-consensus sequence are labeled in red, whereas other residues are labeled in blue. Note that only Leu42 is exposed in the surface. Upper panel: ribbon structure; lower panel: surface structure. (**B**) GST pull-down. Immobilized GST or GST-tagged protein as indicated was incubated with recombinant CRM1 and RanGTP. (**C**) RanGTP-dependent interaction between GST-SOD1^G93A and CRM1. Recombinant CRM1 was precipitated by immobilized GST, GST-SOD1^wt or GST-SOD1^G93A with or without RanGTP. (**D**) Native immunoprecipitation (IP). GFP-SOD1^wt and GFP-SOD1^G93A were each expressed in HEK293T cells. The cell lysates prepared under native condition were immunoprecipitated with either the preimmune serum (control) or antibodies against a peptide containing the NES-like sequence of SOD1 (α-NLP). (**E**) Native IP. HEK293T cells expressing different GFP-tagged SOD1 proteins were lysed under native condition and immunoprecipitated with α-NLP. WCL: whole cell lysates.

G93A, and L144S, but not by wt-SOD1 (*Figure 3B*). Disruption of the NES consensus sequence by creating an L38R mutation in G85R or G93A largely diminished their binding to CRM1 (*Figure 3B*, Lane 7 vs 4, Lane 8 vs 5). Moreover, CRM1-NES-like sequence interaction was dependent on the presence of RanGTP. The interaction was diminished when RanGTP was omitted from the pull-down assay (*Figure 3C*), indicating a receptor-cargo relationship for CRM1-NES-like sequence of SOD1. The exposure of the NES-like sequence in cells was determined by native immunoprecipitation (IP).

To do this, we generated rabbit polyclonal antibodies against the NES-like sequence-containing peptide (NLP). Under native condition, anti-NLP efficiently precipitated GFP-SOD1$^{G93A}$ and six other mutants but not GFP-SOD1$^{wt}$ from lysates prepared from HEK293T cells transiently expressing the respective protein (*Figure 3D,E*). These results indicate that the NES-like sequence of SOD1 is normally buried in the wild-type protein but is exposed in the mutants possibly due to protein misfolding.

## All ALS-linked SOD1 mutants tested assume NES-like sequence-exposed conformation but with different tendencies

The GST pull-down and IP results also suggest that the NES-like sequence is exposed in multiple ALS-linked SOD1 mutants. We then asked whether exposure of the NES-like sequence is a common feature for ALS-linked SOD1 mutants. We tested 17 representative ALS-linked SOD1 mutants, including those deficient in copper binding (H46R, H48R, and H120L) or zinc binding (H80R and D83G) and one that cannot form disulfide bond (C57R) (*Valentine et al., 2005*). These mutants were tagged with GFP and expressed in HeLa cells. GFP-SOD1$^{wt}$ was expressed as a negative control. As described above, exposure of the NES-like sequence can result in predominantly cytoplasmic distribution (nuclear < cytoplasmic: N<C) of some SOD1 mutants (*Figure 1A*). Therefore, we first evaluated the exposure of the NES-like sequence in these GFP-tagged mutants by directly observing GFP localization. We found that cells expressing GFP-tagged SOD1 mutants exhibited either a N<C distribution or a distribution pattern similar to GFP-SOD1$^{wt}$ (wt-SOD1-like). While N<C localization was the predominant pattern found in over half of the mutants tested, this localization pattern was found only in a fraction of cells expressing the remaining mutants, including L38R (0%), H46R (3.9%), C57R (62.6%), H80R (53.7%), D90A (59.5%), H120L (55.5%), L126S (66.5%) and L144S (59.8%) (*Figure 4A* and *Figure 4—figure supplement 1*). Importantly, disruption of the NES-consensus sequence by adding an L38R mutation to eight tested mutants resulted in wt-SOD1-like GFP distribution in all transfected cells (*Figure 4—figure supplement 2*). We then examined the exposure of the NES-like sequence by anti-NLP immunofluorescence. Positive anti-NLP staining was found in 90–100% of cells for all mutants except for L126S (81%) and L144S (77%) (*Figure 4A,B* and *Figure 4—figure supplement 1*). Consistent with the notion that exposure of the NES-like sequence exports misfolded SOD1 to the cytoplasm, anti-NLP immunofluorescence exhibited predominantly cytoplasmic localization, even in cells exhibiting wt-SOD1-like GFP localization, such as those expressing H46R (*Figure 4A,B*). The L38R mutant that has a disrupted NES-consensus sequence, as an exception, was stained both in the nucleus and the cytoplasm (*Figure 4B*). These results suggest that all SOD1 mutants tested expose the NES-like sequence, but the propensities to expose it vary among the mutants. Native IP with anti-NLP antibodies further confirmed that mutants with less N<C GFP distribution (H46R, D90A, L126S and L144S) were also less efficiently precipitated by anti-NLP, compared to mutants showing predominant N<C GFP distribution (A4V, G85R, G93A and G108V) (*Figure 4C*). Interestingly, available reports indicate that ALS patients with H46R, D90A, L126S or L144S mutation have slow disease progression, whereas disease progress faster in patients with A4V, G85R, G93A or G108V mutation (*Bali et al., 2017*; *Byström et al., 2010*). It is possible that SOD1 mutants can assume more than one conformations in cells, among which their tendencies to expose the NES-like sequence correlate with disease progression.

## Misfolded wt-SOD1 exposes the NES-like sequence

We found that about 2% of the cells expressing GFP-SOD1$^{wt}$ were positive for anti-NLP staining, suggesting that the NES-like sequence is exposed in misfolded wt SOD1 (*Figure 4A*). In support for this observation, previous studies have shown that wt-SOD1 can become misfolded under certain stress conditions and the misfolded wt-SOD1 adopts a 'toxic conformation' that is similar to ALS-linked SOD1 mutants (*Rotunno and Bosco, 2013*). We, therefore, further investigated whether the NES-like sequence is exposed in misfolded wt-SOD1. It was reported that zinc binding stabilizes wt-SOD1, whereas chelation of zinc leads SOD1 to adopt a 'mutant-like conformation' (*Homma et al., 2013*). We, therefore, treated HeLa cells expressing GFP-SOD1$^{wt}$ with the zinc chelator, N,N,N′,N′-tetrakis (2-pyridylmethyl) ethylenediamine (TPEN) followed by anti-NLP staining. More cells lost nuclear distribution of GFP-SOD1$^{wt}$ following TPEN treatment, and these cells were specifically stained by anti-NLP (*Figure 4D*). Anti-NLP also stained endogenous SOD1 in TPEN-treated cells,

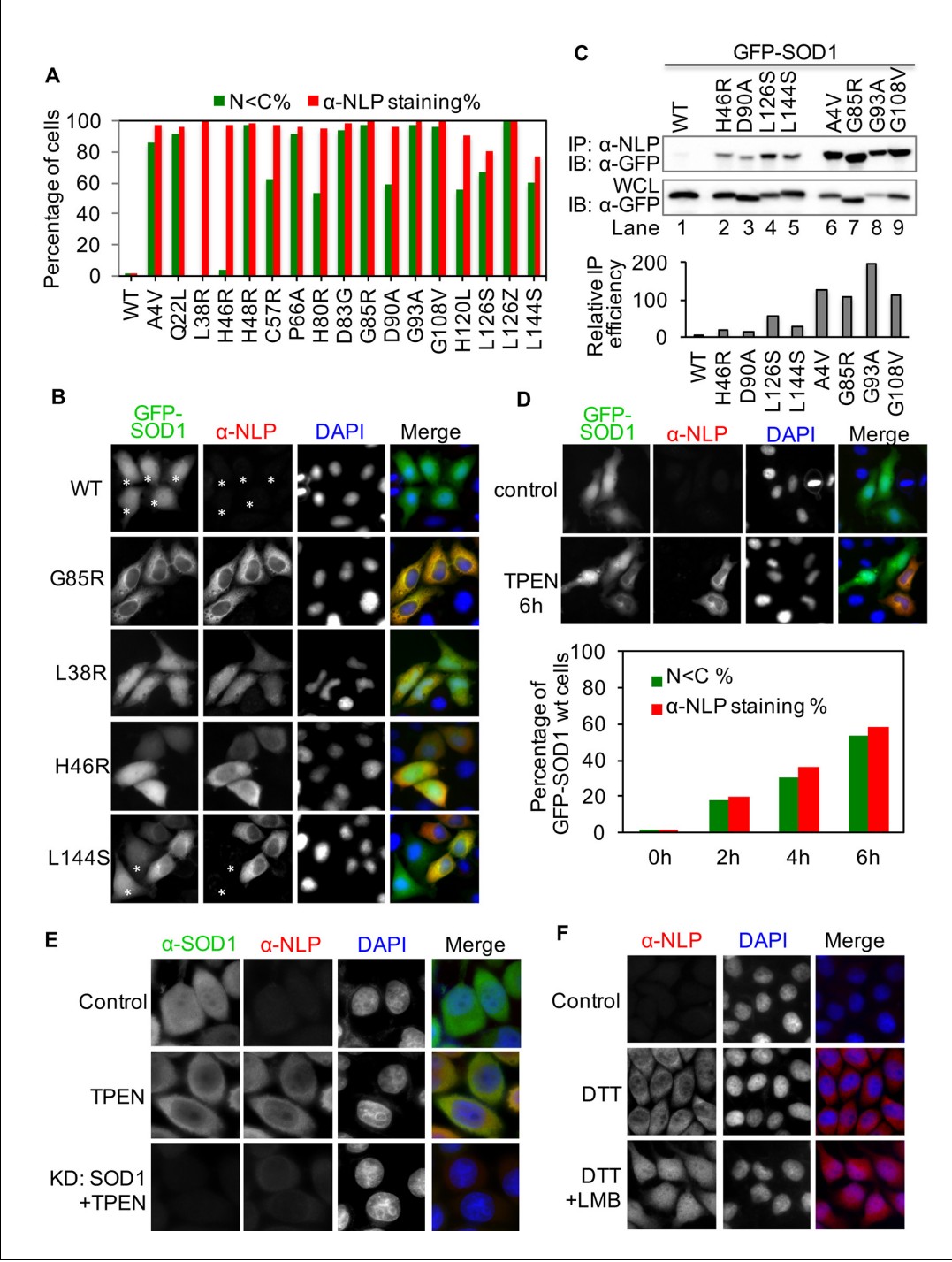

**Figure 4.** The NES-like sequence is exposed in ALS-linked SOD1 mutants and misfolded wt-SOD1. (**A**) HeLa cells expressing GFP-tagged wt-SOD1 or indicated mutants were stained with α-NLP. Percentages of cells exhibiting predominantly cytoplasmic GFP distribution (N<C%) and cells positively stained by α-NLP were plotted. n = 392 (WT), n = 395 (A4V), n = 340 (Q22L), n = 431 (L38R), n = 389 (H46R), n = 411 (H48R), n = 423 (C57R), n = 376 (P66A), n = 393 (H80R), n = 402 (D83G), n = 450 (G85R), n = 435 (D90A), n = 408 (G93A), n = 495 (G108V), n = 355 (H120L), n = 367 (L126S), n = 462 (L126Z), n = 410 (L144S). (**B**) Anti-NLP specifically stained SOD1 mutants but not wt-SOD1 as detected by immunofluorescence microscopy. Asterisks indicate SOD1-expressing cells not being stained by α-NLP. (**C**) Native IP. HEK293T cells expressing different GFP-tagged SOD1 proteins were lysed under native conditions and immunoprecipitated with α-NLP. Immunoprecipitates and whole cell lysates (WCL) were blotted with α-GFP antibody. Relative IP efficiency for each SOD1 protein was calculated as the ratio of the band density

*Figure 4 continued on next page*

*Figure 4 continued*

from IP sample over that from corresponding WCL sample and was plotted as fold-difference relative to wt-SOD1. (D) Demetallation induces exposure of the NES-like sequence in GFP-SOD1$^{wt}$. HeLa cells expressing GFP-SOD1$^{wt}$ were treated with TPEN (10 μM) and then stained with α-NLP. n = 392 (0h), n = 404 (2h), n = 527 (4h), n = 505 (6h). (E) Control HeLa cells or HeLa cells transfected with SOD1 siRNA (KD: SOD1) were treated with TPEN (4 μM) for 20 hr. Then the cells were stained with α-NLP and α-SOD1 antibodies and imaged for immunofluorescence. (F) DTT induces exposure of the NES-like sequence in endogenous SOD1. HeLa cells were treated with DTT (0.4 mM) and LMB (5 nM) as indicated for 20 hr. Then the cells were stained with α-NLP.

The following source data and figure supplements are available for figure 4:

**Source data 1.** Data for *Figure 4A,C and D*.
**Figure supplement 1.** Anti-NLP specifically recognizes various ALS-linked SOD1 mutants but not wt SOD1.
**Figure supplement 2.** A L38R mutation restores the nuclear distribution of various ALS-linked SOD1 mutants.
**Figure supplement 3.** Oxidative stress induces exposure of the NES-like sequence in endogenous SOD1.

whereas a non-conformation-specific anti-SOD1 antibody (71G8) stained both control and TPEN-treated cells (*Figure 4E*). Moreover, in TPEN-treated cells, both 71G8 and anti-NLP showed predominantly cytoplasmic staining (*Figure 4E*), suggesting that misfolded endogenous SOD1 is also exported from the nucleus. The specificity of anti-NLP staining of endogenous SOD1 was demonstrated by siRNA-mediated knockdown of SOD1 expression. RNAi dramatically decreased the staining by both 71G8 and anti-NLP staining (*Figure 4E*). Similarly, cells treated with sodium arsenite, an oxidative stress inducer, were also specifically stained by anti-NLP, and the staining was significantly reduced when SOD1 was knocked down (*Figure 4—figure supplement 3*). Treatment of HeLa cells with dithiothreitol (DTT), a strong reducing agent that may break the disulfide bond in SOD1, also resulted in predominantly cytoplasmic staining of anti-NLP (*Figure 4F*). LMB treatment of DTT-treated cells increased anti-NLP staining in the nucleus (*Figure 4F*), indicating that the reduced nuclear staining is due to CRM1-dependent nuclear export of misfolded SOD1. Taken together, these results suggest that certain environmental stresses can induce wt-SOD1 to adopt 'mutant-like' conformation with exposed NES-like sequence.

## Anti-NLP detects mutant SOD1 in samples from ALS patients

To determine the clinical relevance of this novel conformation, we performed anti-NLP immunostaining of spinal cord sections from human ALS patients. The specificity of the anti-NLP immunohistochemistry was first demonstrated by a lack of staining in spinal cord sections from non-transgenic mice (*Figure 5A*). Conversely, the motor neurons and neurites in the spinal cord sections from SOD1$^{L126Z}$ transgenic mice were anti-NLP positive (*Figure 5B*) (*Deng et al., 2011*). In addition, the staining was predominantly cytoplasmic in most anterior horn neurons, which is consistent with the distribution pattern we have shown in HeLa cells for GFP-SOD1$^{L126Z}$ (*Figure 4—figure supplement 1*). The spinal cord sections from three ALS patients harboring A4V or G85R mutation were also stained with anti-NLP. The anti-NLP staining appeared cytoplasmic, both defuse and as aggregates, in anterior horn neurons and neurites (*Figure 5C–E*).

## Nuclear export of misfolded SOD1 decreases proteotoxicity in the nucleus

Misfolded SOD1 mutants are toxic to neurons. When present in the nuclei of neurons, SOD1 mutants are likely to cause proteotoxicity in the nucleus (*Shibata and Morimoto, 2014*; *Gallagher et al., 2014*). Therefore, nuclear clearance may modulate toxicity of SOD1 mutants through reducing proteotoxicity in the nucleus. To test this possibility, we proposed to restore the nuclear localization of SOD1 mutants by disrupting the NES-like consensus sequence using G85R as an example. As shown earlier in *Figures 2E* and *3A*, Leu42, one of the NES consensus residue in human SOD1, normally exists as Gln (Q) in mouse and is exposed on the SOD1 protein surface.

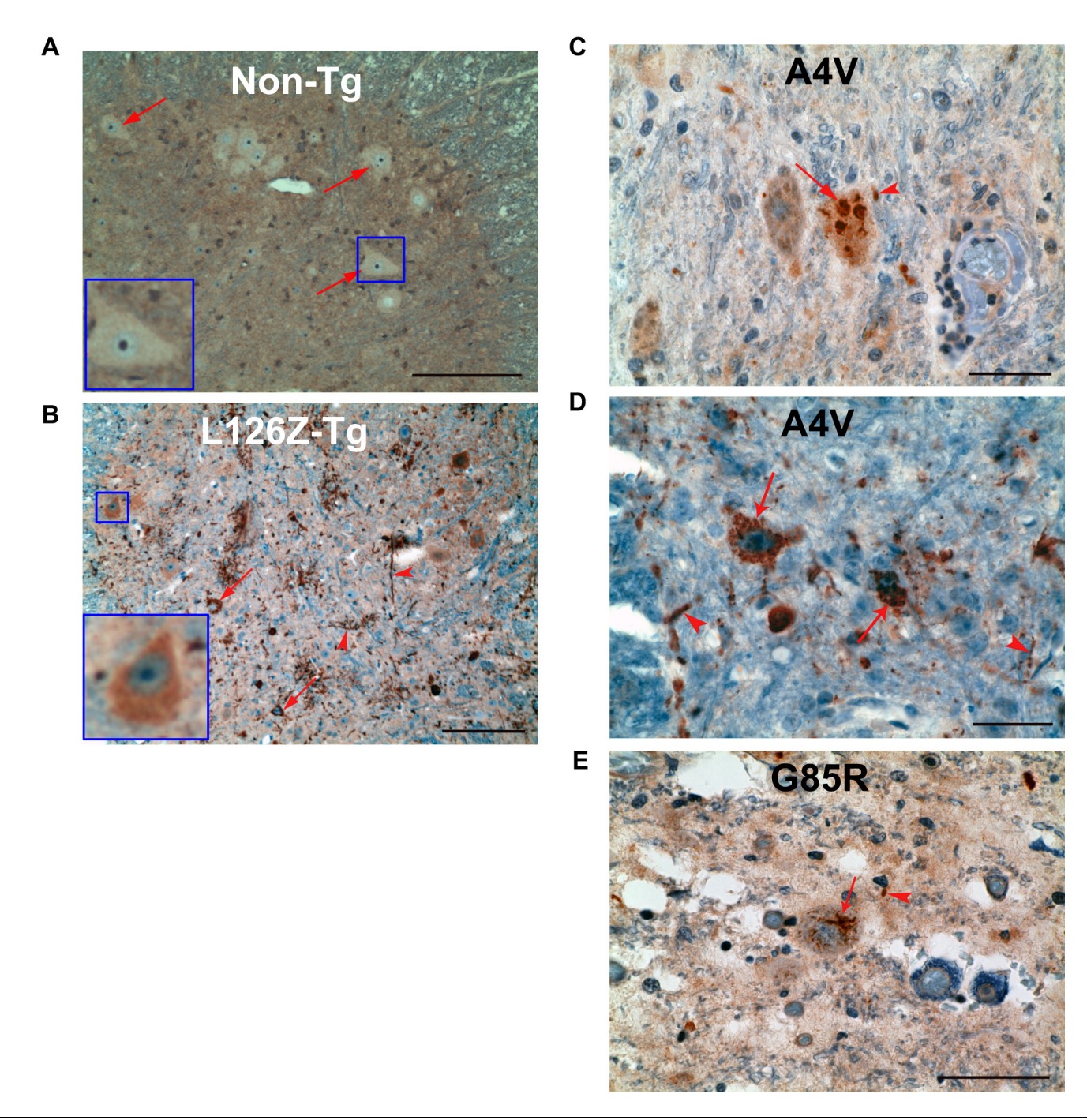

**Figure 5.** Exposure of the NES-like sequence in spinal cord sections from transgenic mice and ALS cases. (**A**). Spinal cord sections from non-transgenic mice were stained with α-NLP. Representative anterior horn neurons are indicated by arrows and also showed in the inset. Scale bar: 100 m. (**B**) Spinal cord sections from *SOD1^L126Z^* transgenic mice were stained with anti-NLP. Representative SOD1 aggregates in anterior horn neurons and neuritis are indicated by arrows and arrowheads, respectively. Inset shows a representative anterior horn neuron with diffused α-NLP staining predominantly in the cytoplasm. Scale bar: 300 m. (**C**) to (**E**) Autopsy spinal cord sections from two SOD1^A4V^ ALS patients (**C** and **D**) and a SOD1^G85R^ ALS patient (**E**) were stained with anti-NLP. Representative SOD1 aggregates in anterior horn neurons and neurites are indicated by arrows and arrowheads, respectively. Scale bar: 100 m.

Therefore, a L42Q substitution is expected to disrupt the NES consensus sequence without affecting SOD1 structure. We generated lentiviral plasmids encoding GFP-SOD1$^{wt}$, GFP-SOD1$^{L42Q}$, GFP-SOD1$^{G85R}$ and GFP-SOD1$^{G85R/L42Q}$, and expressed the variants in NSC34 motor neuron-like cells. As expected, GFP-SOD1$^{wt}$, GFP-SOD1$^{L42Q}$, and GFP-SOD1$^{G85R/L42Q}$ were localized in both the nucleus and the cytoplasm, whereas GFP-SOD1$^{G85R}$ was primarily in the cytoplasm (*Figure 6A*). Importantly, native IP showed that GFP-SOD1$^{L42Q}$ was not recognized by anti-NLP, whereas about same amounts of GFP-SOD1$^{G85R/L42Q}$ and GFP-SOD1$^{G85R}$ were precipitated (*Figure 6B*), supporting that L42Q mutation has little or no effects on SOD1 folding. The cytotoxic effects of GFP-SOD1$^{G85R}$, GFP-SOD1$^{L42Q}$, and GFP-SOD1$^{G85R/L42Q}$ were examined by a WST-1 assay for cell viability, and GFP-SOD1$^{wt}$-expressing cells were used as a control. As previously reported (*Kabashi et al., 2012*; *Kitamura et al., 2014*), proteostress was imposed through inhibition of proteasomal degradation. Viability was similar for cells expressing GFP-SOD1$^{wt}$ and GFP-SOD1$^{L42Q}$ but significantly decreased in cells expressing GFP-SOD1$^{G85R}$ (*Figure 6C*). Cells expressing GFP-SOD1$^{G85R/L42Q}$ showed lower viability than those expressing GFP-SOD1$^{G85R}$, suggesting that disruption of the NES by L42Q mutation increases the cytotoxicity of GFP-SOD1$^{G85R}$ (*Figure 6C*). These results suggest that nuclear export of SOD1 mutants plays a defensive role against proteotoxicity in the nucleus.

We further determined the effects of restoring SOD1$^{G85R}$ nuclear localization in *C. elegans*. A *C. elegans* model of ALS has been established previously by expression of SOD1$^{G85R}$-YFP in neurons (*Wang et al., 2009*). For comparison, we generated new transgenic *C. elegans* lines expressing SOD1$^{L42Q}$-YFP or SOD1$^{G85R/L42Q}$-YFP. Three independent *C. elegans* lines each for SOD1$^{L42Q}$-YFP and SOD1$^{G85R/L42Q}$-YFP were used in this study. As observed in cultured cells, SOD1$^{wt}$-YFP and SOD1$^{L42Q}$-YFP were localized in both the cytoplasm and the nucleus of the neurons throughout the lifespan of *C. elegans* (*Figure 7A,B*). SOD1$^{G85R}$-YFP was cleared from the nuclei of motor neurons in both L1 and adult *C. elegans* (*Figure 7A,B*). In contrast, SOD1$^{G85R/L42Q}$-YFP was localized in both the cytoplasm and the nuclei in motor neurons in L1 *C. elegans* (*Figure 7B*). In adult *C. elegans*, the nuclear localization of SOD1$^{G85R/L42Q}$-YFP was only observed in a fraction of the neurons and no visible aggregate was observed in the nucleus. (*Figure 7B*). A possible explanation is that cytoplasmic aggregation of the mutant protein prevents its entry into the nucleus. Indeed, we found that in adult *C. elegans*, like SOD1$^{G85R}$-YFP, SOD1$^{G85R/L42Q}$-YFP formed large cytoplasmic aggregates (*Figure 7B*). Immunoblotting revealed that these new lines expressed similar levels of the SOD1 transgenes as the previously generated SOD1$^{wt}$-YFP and SOD1$^{G85R}$-YFP lines, with the exception of the line L42Q-1 (*Figure 7C*). Similar amount of SOD1$^{G85R}$-YFP and SOD1$^{G85R/L42Q}$-YFP, but not SOD1$^{wt}$-YFP or SOD1$^{L42Q}$-YFP were detected in the insoluble fractions (*Figure 7C*, upper panel, lanes 2, 6–8), indicating the formation of aggregates for the variants in these *C. elegans* lines. To assess the effects of SOD1$^{G85R}$-YFP and SOD1$^{G85R/L42Q}$-YFP on locomotion, we measured the body bending rates for L4 larvae of all these lines. As controls, the lines expressing SOD1$^{wt}$-YFP and SOD1$^{L42Q}$-YFP showed significantly higher bending rates than SOD1$^{G85R}$-YFP and SOD1$^{G85R/L42Q}$-YFP lines (*Figure 7D*). Importantly, all three SOD1$^{G85R/L42Q}$-YFP lines showed significantly lower bending rates than the SOD1$^{G85R}$-YFP line (*Figure 7D*). Next, we compared the survival rates of the transgenic lines. SOD1$^{L42Q}$-YFP and SOD1$^{wt}$-YFP worms showed similar survival rates, whereas the SOD1$^{G85R}$-YFP line had significantly decreased survival rate, in agreement with previous reports. Remarkably, about 80% of all worms expressing SOD1$^{G85R/L42Q}$-YFP died by day 3 (*Figure 8A,B*). A potential cause for the decreased survival of SOD1$^{G85R/L42Q}$-YFP worms was identified, as these animals exhibited an egg-laying defect where eggs hatched inside mothers, a phenotype known as 'bagging'. About 70% of SOD1$^{G85R/L42Q}$-YFP worms showed the bagging phenotype, with an average of 4.2 eggs hatched per mother. This was significantly worse than observed for SOD1$^{G85R}$-YFP animals, where about 17% of worms showed a bagging phenotype, with an average of 2.4 eggs hatched per mother (*Figure 9A,B,C*). Remarkably, less eggs were laid by SOD1$^{G85R/L42Q}$-YFP animals in 48 hr compared to other lines (*Figure 9D*). These results support increased toxicity of SOD1$^{G85R/L42Q}$-YFP compared to SOD1$^{G85R}$-YFP when expressed in neurons of *C. elegans*.

Taken together, results from the cell and the animal models indicate that accumulation of misfolded SOD1 in both the cytoplasm and nucleus is more toxic than accumulation only in the cytoplasm. Therefore, removal of the misfolded SOD1 from the nucleus, mediated by the exposed NES-like sequence and CRM1-dependent nuclear export, is potentially a defense mechanism against the proteotoxic effects of ALS-linked SOD1 mutants in the nucleus.

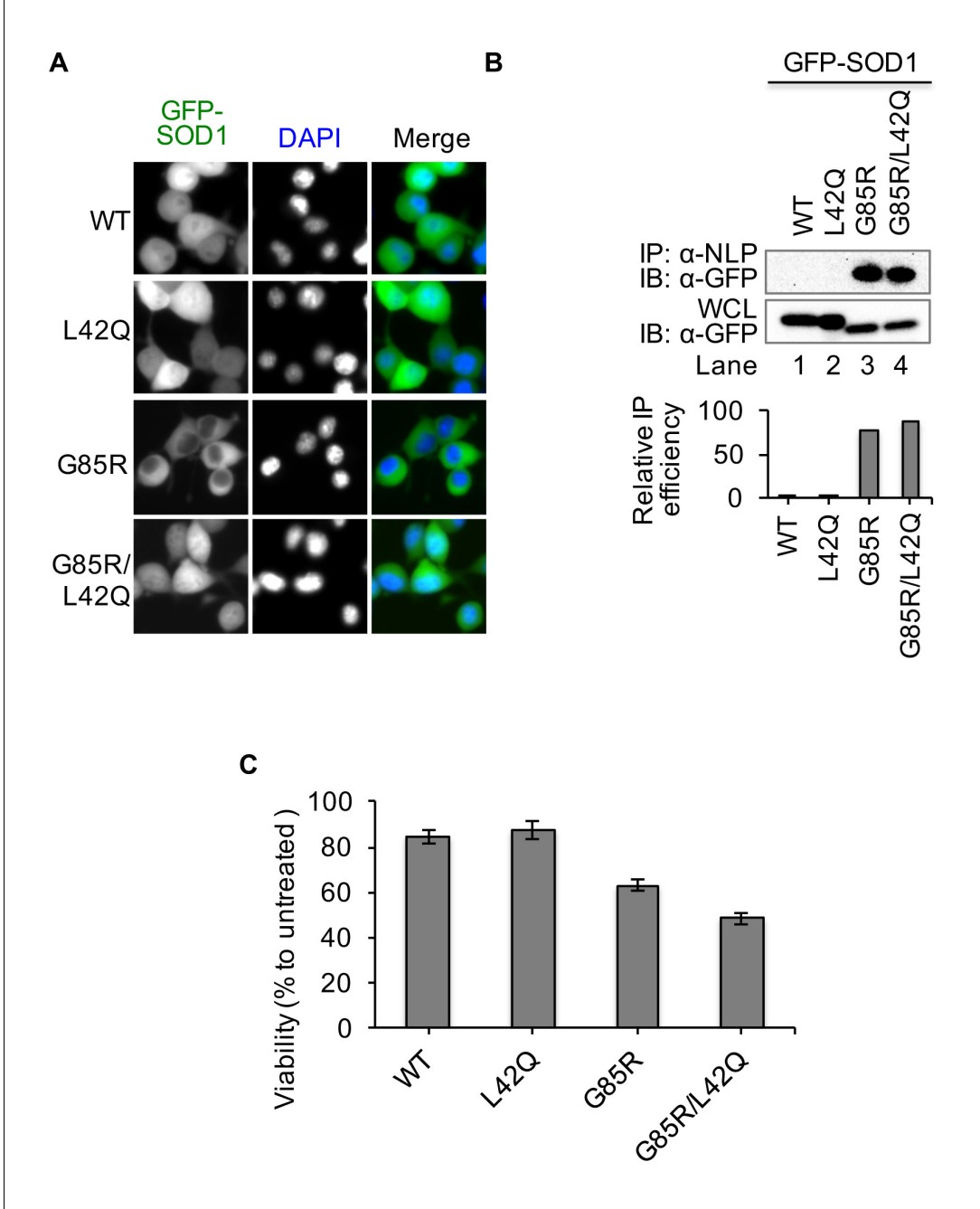

**Figure 6.** Disruption of the NES by L42Q mutation in SOD1[G85R] mutant results in higher cytotoxicity in NSC34 cells. NSC34 cells were infected with lentiviruses expressing GFP-tagged WT, L42Q, G85R or G85R/L42Q SOD1 proteins. (**A**) SOD1[L42Q] and GFP-SOD1[G85R/L42Q] have similar subcellular localizations as GFP-SOD1[wt] in NSC34 cells. (**B**) Native IP. NSC34 cells expressing different GFP-tagged SOD1 proteins were lysed under native condition and immunoprecipitated with α-NLP. Immunoprecipitates and whole cell lysates (WCL) were blotted with α-GFP antibody. Relative IP efficiency for each SOD1 protein was calculated as the ratio of the band density from IP sample over that from corresponding WCL sample and was plotted as fold-difference relative to wt-SOD1. (**C**) Cell viability (WST-1 assay). NSC34 cells expressing GFP-tagged SOD1 proteins as indicated were treated with MG132 (5 μM) for 24 hr. Data represent means ± SEM, n = 4. Unpaired *t*-test, G85R/L42Q vs G85R: p=0.005; WT vs L42Q: p=0.564.

The following source data is available for figure 6:

**Source data 1.** Data for *Figure 6B and C*.

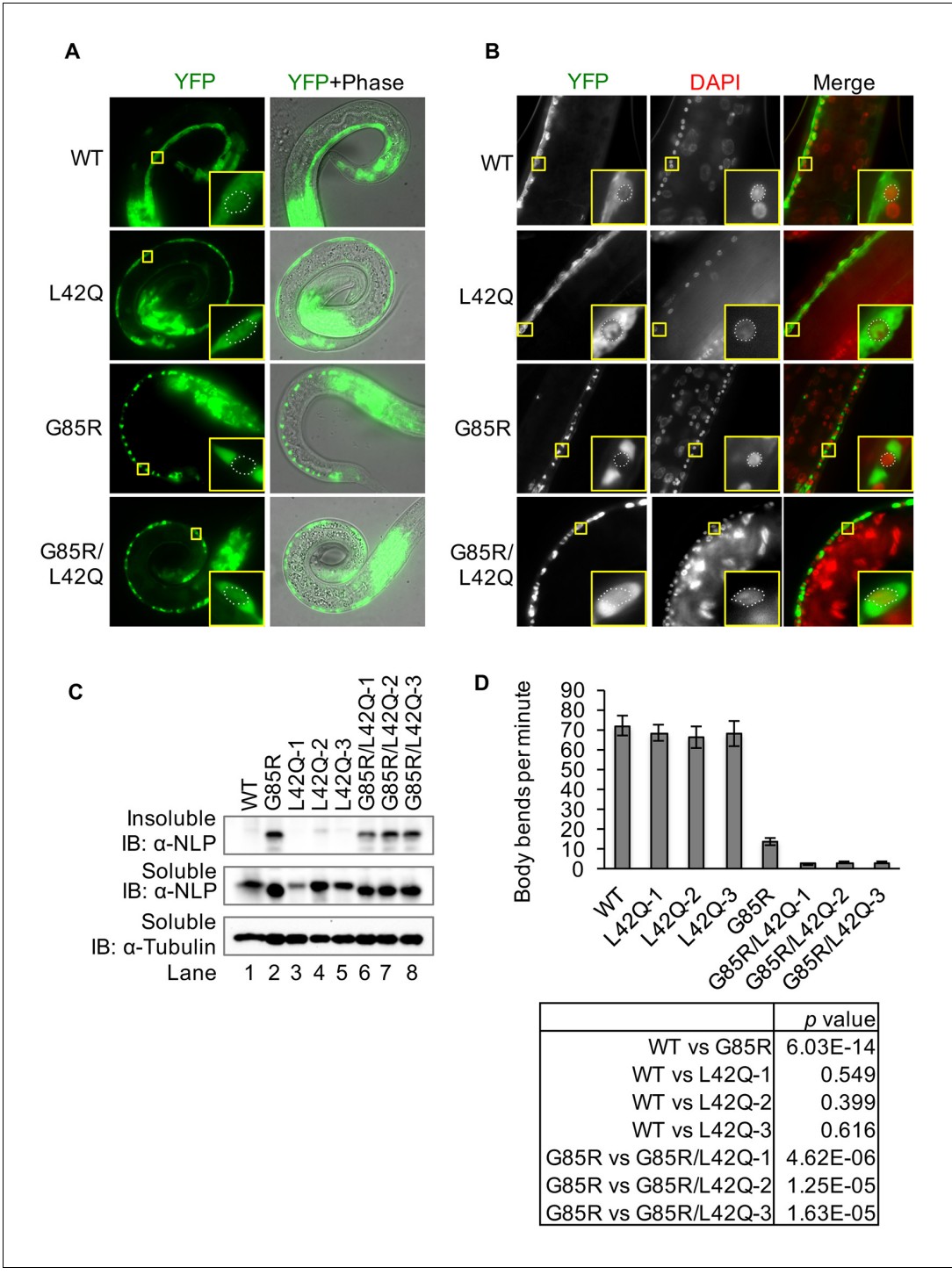

**Figure 7.** Disruption of the NES by L42Q mutation in SOD1$^{G85R}$ mutant causes severe defects in locomotion in transgenic *C. elegans*. Transgenic worm lines neuronal specifically expressing SOD1$^{wt}$-YFP (WT), SOD1$^{L42Q}$-YFP (L42Q), SOD1$^{G85R}$-YFP (G85R) or SOD1$^{G85R/L42Q}$-YFP (G85R/L42Q) in neurons were generated previously or in this study. (**A**), Expression of YFP-tagged SOD1 proteins in L1 of transgenic worm lines. Insets show higher magnification of motor neurons. The dot lines depict the nuclear profiles of the enlarged neurons (also in B). (**B**) Microscopy of adult animals. DAPI (pseudo colored red) was used to stain the nucleus. (**C**) Immunoblotting. L1 larvae were lysed by sonication on ice. The soluble (S) and insoluble (P) fractions were processed for immunoblotting. (**D**) Body bending rates. L4 animals were transferred to a drop of M9 buffer and counted for their total body bends in 1 min. Data represent means ±SEM, n = 20. Two-tailed unpaired t-test was used to calculate the p values.

*Figure 7 continued on next page*

*Figure 7 continued*

The following source data is available for figure 7:

**Source data 1.** Data for *Figure 7D*.

## Discussion

In this study, we found that SOD1 misfolding caused by either mutation or chemical insults induces nuclear clearance of SOD1 protein. The underlying mechanism is the exposure of a normally buried NES-like sequence caused by the misfolding event. CRM1 can recognize the NES-like sequence and facilitates export of the misfolded SOD1 from the nucleus to the cytoplasm. Exposure of the NES-like sequence is a common conformational feature for all 17 representative ALS-linked SOD1 mutants tested, and also for misfolded wt-SOD1. Moreover, this conformational feature is also present in spinal cord motor neurons of human ALS patients. Our data also suggest that the exposure of the NES-like sequence is a disease-modifying feature for SOD1 mutants. First, this conformation appears to correlate with ALS disease progression. Several mutants that show lower propensities to acquire the NES-like sequence-exposed conformation, including H46R, D90A, L126S and L144S, have been previously reported to associate with slow progression of ALS (*Bali et al., 2017*; *Byström et al., 2010*). Second, although the potential toxic effects of this exposed NES-like sequence itself were not evaluated by the present study, data from cells and *C. elegans* suggest that nuclear export of misfolded SOD1 decreases the general toxicity of G85R mutant, probably due to diminishing the proteotoxicity in the nucleus. Disruption of the NES-consensus in SOD1$^{G85R}$ significantly increases defects in locomotion and egg-laying in *C. elegans* models. It is likely that loss of muscle function as a result of more severe motor neuron degeneration is responsible for these severe phenotypes. Therefore, the ability of neurons to remove misfolded proteins from the nucleus after exposure of the NES-like sequence may play an important role in pathogenesis of SOD1-linked ALS.

Nuclear export not only decreases the levels of SOD1 mutants in the nucleus, but also increases their levels in the cytoplasm. Loss of nuclear SOD1 may lead to loss of function in the nucleus, but it is unlikely a disease mechanism because SOD1 knockout does not lead to ALS phenotype and pathology (*Reaume et al., 1996*). To the contrary, knockdown of SOD1 mutants by siRNA are currently being pursued as a therapeutic approach for ALS (*Ralph et al., 2005*; *Thomsen et al., 2014*; *van Zundert and Brown, 2017*). Nuclear export may facilitate the removal of misfolded SOD1, which would produce beneficial effects. It is known that the nucleus relies on exclusively the proteasomal degradation pathway to remove misfolded and unwanted proteins, whereas the cytoplasm has both proteasomal and lysosome-dependent autophagy pathways. Thus, nuclear export may facilitate mutant SOD1 degradation by autophagy. On the other hand, nuclear export increases the cytoplasmic concentration of the abnormal SOD1, which may promote toxic effects on the functions of cytoplasmic organelles, for example, ER and mitochondria. It is likely that NES-like sequence-mediated changes in subcellular localization of SOD1 mutants would have complex effects on neuronal functions and survival, but collectively, may reduce the toxicity of SOD1 mutants. Future careful interrogation of this complex issue is warranted.

Our results also indicate that exposure of the NES-like sequence may be a common property of misfolded SOD1. Although we did not test every previously reported ALS-linked SOD1 mutations, those examined span almost the full length of SOD1 and also included almost all representative mutants. Misfolded wt-SOD1 also shares this feature, which supports the controversial proposition that misfolding of wt-SOD1 is involved in pathogenesis of sporadic ALS (*Rotunno and Bosco, 2013*; *van Zundert and Brown, 2017*; *Ayers et al., 2016*; *Grad et al., 2014*). The exposure of NES-like sequence can be specifically detected with anti-NLP antibody, which may become a useful tool for research and diagnosis. Interestingly, several other antibodies that recognize regions covered or partially overlapping with the NES-like sequence have been reported, including D3H5 (epitope: residues 24–55) (*Rotunno and Bosco, 2013*; *Gros-Louis et al., 2010*), AJ10 (epitope: residues 29–57) (*Sábado et al., 2013*), and USOD (epitope: residues 42–48) (*Kerman et al., 2010*). All these antibodies were also reported to selectively recognize mutant SOD1 over wt-SOD1, suggesting that the

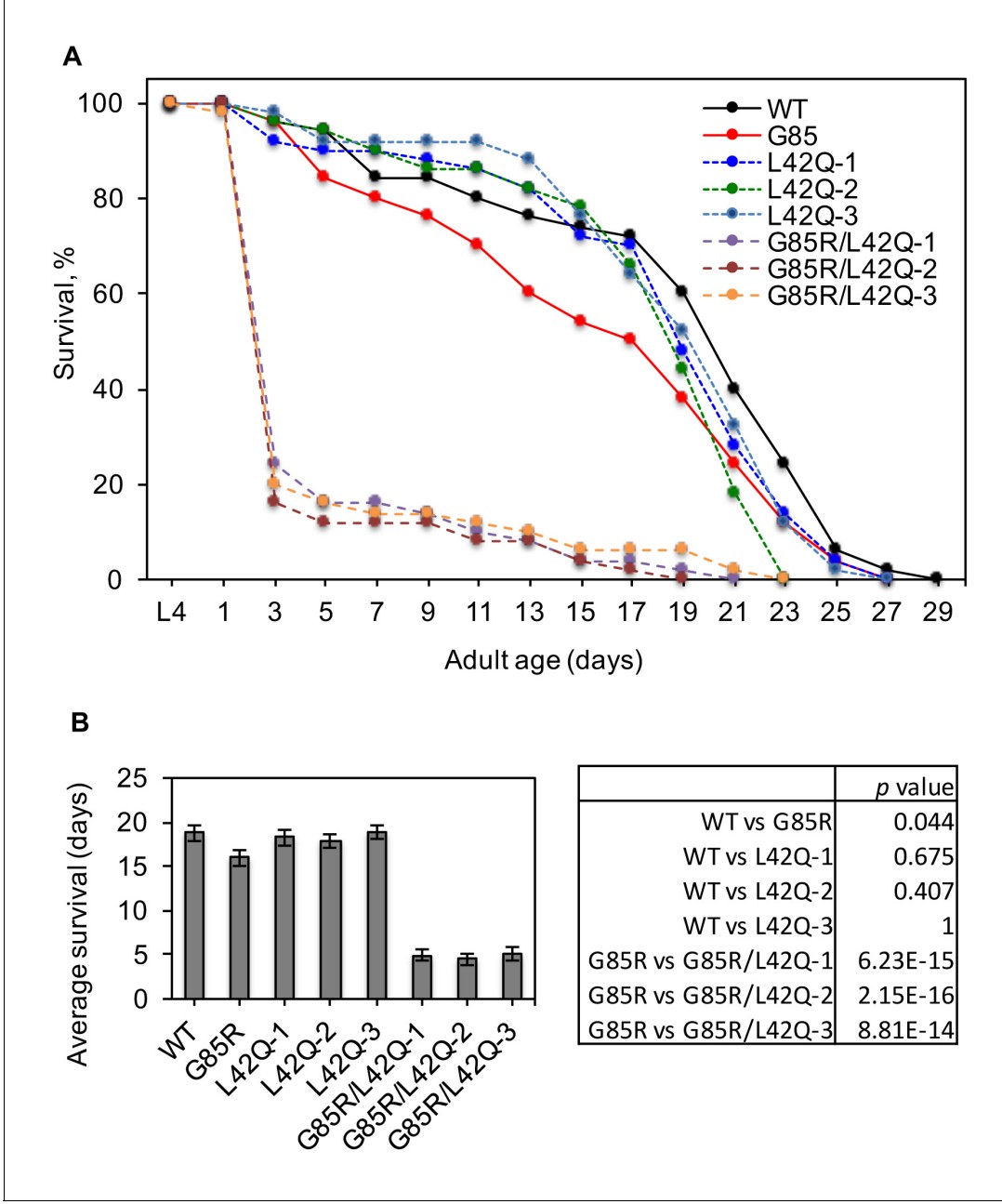

**Figure 8.** Disruption of the NES by L42Q mutation in SOD1[G85R] mutant decreases survival of transgenic *C. elegans*. (A) Survival curves. Mid-L4 animals were picked and followed for their survival. The surviving worms were transferred to fresh plates every 2 days until all the worms died. n = 50 for each transgenic line. (B) Average survival days from (A). Two-tailed unpaired t-test was used to calculate the p values.

The following source data is available for figure 8:

**Source data 1.** Data for *Figure 8A and B*.

epitopes formed by the NES-like sequence are highly specific for misfolded SOD1. Therefore, the NES-like sequence may serve as a biomarker for the detection of misfolded SOD1 in ALS patients. Moreover, this immunogenic region may be used for immunotherapy for SOD1-linked ALS as was reported for the SOD1 exposed dimer interface (SEDI) peptide, that is specifically exposed in monomeric SOD1 (*Rakhit et al., 2007*; *Liu et al., 2012*).

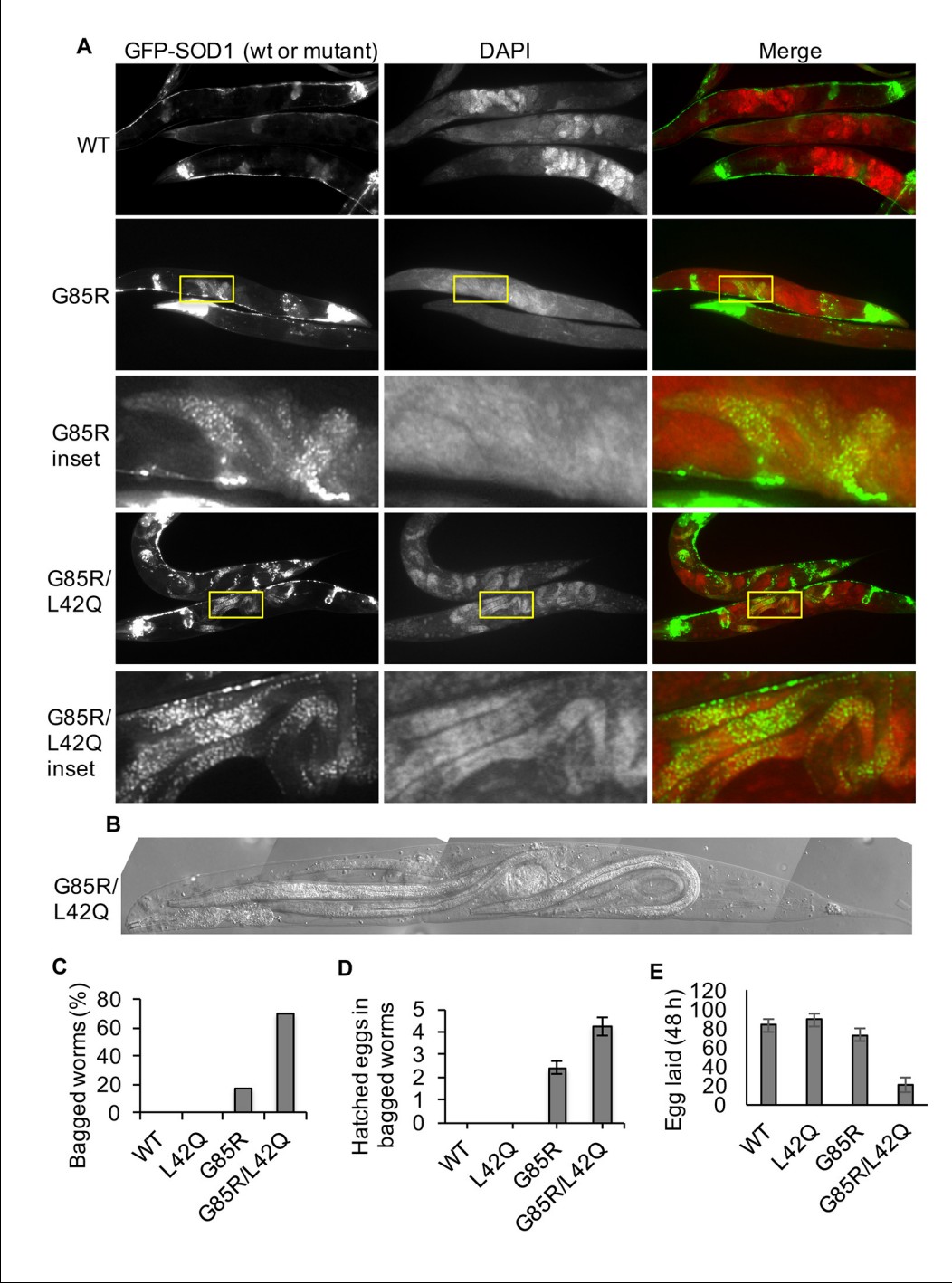

**Figure 9.** Disruption of the NES by L42Q mutation in SOD1^G85R mutant causes severer egg-laying defect in transgenic *C. elegans*. (A) Bagging in transgenic worms. Insets show hatched eggs in bagged worms. (B) An example of bagged G85R/L42Q *C. elegans*. (C) Rates of bagging in transgenic worms. L4 animals were transferred to fresh plates, and bagged worms were counted 48 hr later. n = 46 (WT), n = 45 (L42Q), n = 41 (G85R), n = 47 (G85R/L42Q). (D) Average hatched eggs in bagged worms from (C). n = 7 (G85R), n = 33 (G85R/L42Q). Data represent means±SEM. G85R/L42Q vs G85R: p=0.048 by two-tailed unpaired t-test. (E) Eggs laid in 48 hr. Each L4 animal was transferred to one well of 96-well dish containing S-complete medium with OP50. 48 hr later, larvae (hatched eggs) and unhatched eggs were counted for each well. Data represent means ±SEM. n = 12. G85R vs. G85R/L42Q: p=4.07E-05 by Two-tailed unpaired *t*-test.

*Figure 9 continued on next page*

*Figure 9 continued*

The following source data is available for figure 9:

**Source data 1.** Data for *Figure 9C,D and E*.

Our findings also have far-reaching implications for nuclear protein quality control. Proper mainte-nance of nuclear protein homeostasis is important for preserving cell function (*Shibata and Mori-moto, 2014*; *Gallagher et al., 2014*). Accumulation of misfolded proteins in the nucleus can affect cell function and cause diseases. A number of neurodegenerative diseases are associated with aggregates and inclusions formed by misfolded proteins in the nucleus, such as polyglutamine (polyQ) repeat diseases, polyalanine (polyA) repeat diseases, the RNA-mediated diseases, neuronal intranuclear inclusion disease (NIID), neuronal intermediate filament inclusion disease (NIFID), multi-ple system atrophy (MSA), neuroferritinopathy, and inclusion body myopathy with early onset Paget's disease and frontotemporal dementia (IBMPFD) (*Gallagher et al., 2014*; *Woulfe, 2008*). In addition to the known protein quality control mechanisms, such as chaperones and ubiquitin-protea-some system (*Shibata and Morimoto, 2014*; *Gallagher et al., 2014*), our findings suggest that exposure of a normally buried NES-like sequence leading to CRM1-dependent nuclear export of the misfolded SOD1 may represent a novel mechanism for maintaining nuclear proteostasis. Exposure of normally buried hydrophobic regions is a common feature of misfolded proteins. These hydrophobic regions are commonly recognized by chaperones and quality control E3 ubiquitin ligases, to initiate the refolding and ubiquitination of the misfolded proteins, respectively (*Rosenbaum et al., 2011*; *Fredrickson et al., 2013*; *Horwich, 2014*). Interestingly, proteomic analysis has revealed that NES consensus sequences are frequently observed in hydrophobic regions buried inside of proteins (*Xu et al., 2012*). This suggests that many misfolded proteins may expose buried hydrophobic regions containing NES consensus sequences, which may result in being exported from the nucleus by CRM1 if not efficiently degraded by the nuclear proteasomes or refolded by chaperones. Nuclear export may be a mechanism responsible for cytosolic mislocalization of many disease-causing mutant proteins (*Hung and Link, 2011*). Notably, failure in nuclear import is also a cause for mislocalization of some disease proteins. For example, mutations in the NLS of FUS, which also cause ALS, leads to its mislocalization to the cytoplasm (*Dormann et al., 2010*; *Gal et al., 2011*). Mutations in another ALS-linked gene, TDP43, also lead to exclusion of TDP43 protein from the nucleus of the neurons and the formation of cytoplasmic TDP43 aggregates, but the underlying mechanism is not clear (*Mackenzie et al., 2007*; *Deng et al., 2010*). Abnormal nucleocytoplasmic transport may represent a convergent pathogenic pathway for some neurodegenerative diseases. Most recent studies have suggested that the nuclear-cytoplasmic transport is disrupted by the hexanucleotide repeat expan-sion (HRE) GGGGCC (G4C2) in C9orf72, which causes ALS and frontotemporal dementia (FTD) (*Zhang et al., 2015*; *Freibaum et al., 2015*; *Jovičić et al., 2015*). It is possible that the exposure of NES-like sequence in mutant and misfolded SOD1 may also affect the nucleocytoplasmic transport of other cargo, because CRM1-mediated export is saturable and SOD1 is a highly abundant protein in the cell. However, there is an important distinction that nuclear export is deleterious in C9orf72-caused ALS, but it is protective in SOD1-linked ALS.

## Materials and methods

### Plasmid constructs

pAcGFP1-SOD1-WT, -A4V, -G93A were acquired from Addgene (*Stevens et al., 2010*). All other mutants used were created by site-directed mutagenesis. To express HIV Rev1 NES, P1 or P2 pep-tide of SOD1 in a fusion with mCherry, DNA fragments encoding these peptides were individually inserted into Bgl II/ Sal I sites of pmCherry-C1. To express recombinant GST-tagged SOD1 proteins or peptides, Bgl II/Sal I fragments encoding SOD1 wt or mutants, P1 or P2 peptide of SOD1, or PKI NES peptide were individually inserted into BamH I/Sal I sites of pGEX-4T-3. pET3a-CRM1 was kindly provided by Dr. Dirk Görlich (*Paraskeva et al., 1999*). Nde I/BamH I fragment of pET3a-CRM1 encoding CRM1 was inserted into Nde I/BamH I sites of pET28a(+), to create pET28a-CRM1

expressing N-terminal His-tagged CRM1. pcDNA3-FLAG-Ran WT and Q69L were kindly provided by Dr. Mien-Chie Hung (*Giri et al., 2005*).

### Cell culture

HEK293T, HeLa, and NSC34 cells were obtained from ATCC. These cells were not independently authenticated. Mycoplasma contamination is monitored frequently in our laboratory by cytoplasmic DAPI staining and by PCR. All cells were cultured in DMEM (Dulbecco's modified Eagle's medium) supplemented with 10%(V/V) fetal bovine serum, 50 units/ml penicillin, 50 µg/ml streptomycin and 2 mM Glutamine under 5% $CO_2$ in a humidified incubator.

### Antibodies

Mouse monoclonal anti-CRM1 (C-1, sc-74454) antibody was purchased from Santa Cruz Biotechnology (Dallas, TX). Mouse monoclonal anti-FLAG (M2, F3165), anti-$\alpha$-Tubulin (B-5-1-2) and anti-$\beta$-Actin (AC-74) antibodies were purchased from Sigma-Aldrich (St Louis, MO). Mouse monoclonal anti-SOD1 (71G8, #4266) antibody was purchased from Cell Signaling Technology (Danvers, MA), respectively. HRP-conjugated anti-GFP (MA5-15256-HRP) was purchased from Thermo scientific (Waltham, MA).

### Polyclonal anti-NLP antibodies

To produce antibodies against the NES like sequence of SOD1, a SOD1 peptide corresponding to residues 33–51 (GSIKGLTEGLHGFHVHEFGC) was synthesized by Peptide 2.0 Inc. (Chantilly, VA). A Cysteine was added to the C-terminus for conjugation of the peptide to keyhole limpet hemocyanin (KLH) to improve the immunogenicity. Anti-NLP was purified from rabbit antiserum using the same peptide that was biotinylated at the C-terminus and bound to streptavidin magnetic beads (Thermo scientific, Waltham, MA).

### Lentivirus production

plv-AcGFP1-SOD1(wt) was acquired from Addgene. plv-AcGFP1-SOD1-L42Q, G85R and G85R/L42Q were created by site-directed mutagenesis. These backbone plasmids were each co-transfected with helper vectors pDelta8.7 and pVSVG in a ratio of 5:5:3 to HEK293T cells to produce lentiviral particle as previously described (*Zhong and Fang, 2012*).

### RNA interference

Negative control #1, Crm1 and SOD1 siRNAs (SOD266) were previously reported (*Zhong and Fang, 2012*; *Strunze et al., 2005*; *Maxwell et al., 2004*). siRNAs were transfected with Lipofectamine RNAiMAX (Invitrogen, Carlsbad, CA). 48 hr after siRNA transfection, cells were transfected with different plasmids as indicated with Lipofectamine 2000 (Invitrogen, Carlsbad, CA).

### Recombinant proteins

His-tagged RanGTP was prepared as reported (*Zhong et al., 2011*). His-tagged CRM1 was expressed from BL21 (DE3) transformed with pET28a-CRM1. The bacteria were cultured at 30°C and induced with 0.1 mM Isopropyl $\beta$-D-1-thiogalactopyranoside (IPTG) for 1 hr. Then the bacteria were lysed by sonication in 50 mM Tris-HCl pH7.4, 100 mM NaCl, 250 mM Sucrose. His-CRM1 was purified using Ni-NTA-agarose (Qiagen, Germantown, MD) following the user's manual. GST and GST-SOD1 wt and mutants were expressed from JM109 transformed with pGEX-4T-3 or pGEX-SOD1 constructs, respectively. The bacteria were cultured at 37°C and induced with 0.1 mM IPTG for 1 hr and then lysed by sonication in 20 mM Tris-HCl pH7.4, 50 mM NaCl, 0.5% Triton X-100.

### GST pull-down

GST proteins were immobilized on Glutathione-Sepharose 4B beads (Amersham Biosciences, Pittsburgh, PA). Then pre-incubated CRM1 and RanGTP were added to the beads and incubated for 1 hr at room temperature in CRM1 pull-down buffer (20 mM Tris-HCl pH7.4, 50 mM NaCl, 0.5 mM GTP, and 0.15% Nonidet P-40). After washing three times, the beads were boiled in SDS sample buffer and processed for SDS-PAGE and immunoblotting.

## Immunofluorescence and microscopy

For microscopy, the cells were grown on chambered coverglass and stained with a cell permeable nuclear dye Hoechst 33342. When required, the cells were fixed with 2% paraformaldehyde for 5 min at room temperature before microscopy. For immunofluorescence staining, the cells were fixed in 3.7% paraformaldehyde for 30 min and permeabilized in 0.25% Triton X-100 for 5 min. After blocking in 5% bovine serum albumin (BSA) for 30 min, the cells were incubated with primary antibodies as indicated for 1 hr and then labeled with Alexa Fluor 488 or 594 conjugated secondary antibodies for 1 hr. DAPI or Hoechst 33342 was used for nuclear staining. Fluorescent microscopy was performed using a Zeiss Axiovert 200M fluorescent microscope.

## Time-lapse imaging

HeLa cells stably expressing GFP-SOD1$^{G85R}$ were treated with cycloheximide (CHX, 100 nM), in combination with MG132 (30 μM) or LMB (20 nM) as indicated. Live cell images were acquired under a 40x objective lens mounted on a Nikon Eclipse Ti fluorescence microscope equipped with Perfect Focus System, a high-sensitivity CCD camera (QuantEM 512SC; Photometrics, Tucson, AZ), and environment control units. At least 30 cells for each group were selected for time-lapse imaging.

## Immunoprecipitation

Cells were lysed in cell lysis buffer (150 mM NaCl, 10 mM Tris-HCl pH7.4, 1 mM EDTA, 1 mM EGTA, 0.2% Nonidet P-40 and protease inhibitor cocktail). Cell lysate was incubated with antibodies as indicated and protein A Sepharose beads (Zymed) for 2 hr at 4℃. The beads were washed three times with cell lysis buffer and processed for immunoblotting.

## Immunohistochemistry

Immunohistochemistry was performed as previously described (*Deng et al., 2006*, *Deng et al., 2010*). Briefly, 6 μm sections were cut from formalin-fixed, paraffin-embedded brain and spinal cord from mouse model or ALS patients with SOD1 mutations. The sections were deparaffinized and rehydrated by passing the slides in serial solutions as described. After antigen retrieval and blocking, the sections were incubated with affinity-purified anti-NLP rabbit polyclonal antibodies for 1 hr at room temperature and then with a biotinylated secondary antibody for 30 min. Positive signals were developed by first incubating the slides with peroxidase-conjugated streptavidin (BioGenex, San Ramon, CA) and then with 3-amino-9-ethylcarbazole chromogen (BioGenex, Fremont, CA). The slides were examined and photographed under light microscope.

## Cytotoxicity assays

NSC34 cells expressing GFP-tagged SOD1 proteins were seeded to 96-well plates at $2 \times 10^4$ cells/well. The next day, cells were treated with MG132 (5 μM) for 24 hr. The cell viability was determined using Cell Proliferation Reagent WST-1 (Roche, South San Francisco, CA) according to the manufacturer's protocol.

## Caenorhabditis elegans

Stable transgenic N2 Bristol *C. elegans* lines expressing SOD1$^{wt}$-YFP or SOD1$^{G85R}$-YFP were previously reported (*Wang et al., 2009*). Psnb1:L42Q-YFP and Psnb1:G85R/L42Q-YFP were created by site-directed mutagenesis to express SOD1$^{L42Q}$-YFP and SOD1$^{G85R/L42Q}$-YFP in *C. elegans*, respectively (*Wang et al., 2009*). N2 Bristol strain of *C. elegans* was transformed by DNA injection and the extrachromosomal lines were further treated with gamma ray to generate integrated lines stably expressing SOD1$^{L42Q}$-YFP or SOD1$^{G85R/L42Q}$-YFP. Three independent integrants for each construct showing 100% transmission were used for further experiments. To analyze the expression levels and solubility of SOD1 proteins, L1 larvae were suspended in extraction buffer (PBS, 1 mM EDTA, 1 mM EGTA, 1 mM TCEP) with protease inhibitor cocktail (Sigma, St. Louis, MO) and lysed by sonication on ice. The large debris was removed by spinning at 1000 ×g for 5 min. Then the supernatant was centrifuged at 55,000 rpm (100,000×g) for 15 min to separate into soluble (supernatant) and insoluble (pellet) fractions. The insoluble fractions were washed once with the extraction buffer and then solubilized in 4% SDS. Equal amount of both fractions were processed for immunoblotting. The body bending rates and survival curves of transgenic worm lines were analyzed as previously

reported (*Wang et al., 2009*). Briefly, L4 animals were transferred to a drop of M9 buffer and counted for their total body bends in 1 min. To analyze the survival curves of transgenic animals, mid-L4 animals were picked and the surviving worms were transferred to fresh plates every 2 days until all the worms died. For microscopy of the worms, L1 or adult worms were frozen in liquid nitrogen for 10 s to crack the cuticles and then fixed in 3.7% PFA for 30 min. After permeabilization in 0.1% Triton X-100 for 5 min, the worms were stained for nucleus with DAPI for 5 min.

### Statistical analysis

All statistical analyses were performed using GraphPad Prism 5.0 or Microsoft Excel software. Statistical significance was assessed using paired or unpaired two-tailed Student's *t*-test, as indicated in the corresponding figure legends. For all analyses, $p < 0.05$ was considered statistically significant. Reported sample sizes indicate the number of biological replicates, with data obtained from individual cells, organisms, or sample wells. All collected data were included in analyses (no outlier removal). For experiments analyzing cytotoxicity, sample units (*n*) are the number of biological replicates. For all other experiments, *n* represents the number of cells or animals used in the analysis for each group. No statistical method was used to predetermine sample size. However, the sample sizes we used were similar to those reported in previous studies. No samples or animals were excluded from the analyses.

## Acknowledgements

We would like to thank Drs. Mervyn J Monteiro and Ilia V Baskakov for suggestions and discussions during the course of this study. This work is supported by grants from NSF and NIH/NIAAA to SF.

## Additional information

### Funding

| Funder | Grant reference number | Author |
|---|---|---|
| National Science Foundation | 1120833 | Shengyun Fang |
| National Institute on Alcohol Abuse and Alcoholism | R21AA024245 | Shengyun Fang |

The funders had no role in study design, data collection and interpretation, or the decision to submit the work for publication.

### Author contributions

YZ, Data curation, Software, Formal analysis, Validation, Investigation, Visualization, Methodology, Writing—review and editing; JW, Resources, Supervision, Methodology, Writing—review and editing; MJH, Data curation, Investigation, Writing—review and editing; PY, Resources, Supervision, Writing—review and editing; BMH, Validation, Visualization, Methodology; TS, Resources, Supervision, Methodology; BEV, Resources, Validation, Investigation, Visualization, Methodology; H-XD, Resources, Investigation, Visualization, Methodology, Writing—review and editing; SF, Conceptualization, Resources, Data curation, Formal analysis, Supervision, Funding acquisition, Validation, Investigation, Visualization, Methodology, Writing—original draft, Project administration, Writing—review and editing

### Author ORCIDs

Shengyun Fang, http://orcid.org/0000-0001-7280-5463

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
