## [Decision Letter]

Thank you for submitting your article "Nuclear export of misfolded SOD1 mediated by a normally buried NES-like sequence reduces proteotoxicity in the nucleus" for consideration by *eLife*. Your article has been favorably evaluated by Randy Schekman as the Senior Editor and three reviewers, one of whom is a member of our Board of Reviewing Editors. The reviewers have opted to remain anonymous.

The reviewers have discussed the reviews with one another and the Reviewing Editor has drafted this decision to help you prepare a revised submission The requests for additional experiments may prove challenging so please let us know if you believe you can complete this work within a reasonable length of time.

Summary:

This study reports the observation that a wide range of ALS-associated mutations in SOD1 show diminished nuclear localization when over-expressed as GFP-tagged proteins in cultured mammalian cells. The basis for this altered localization is traced to increased nuclear export due to promiscuous interaction with the export factor Crm1 via a hydrophobic patch in SOD1 that is ordinarily buried in the folded protein. Antibodies against this region preferentially recognize mutant protein (or wild type protein under mis-folding conditions), supporting the idea that a wide range of mutants all partially destabilize SOD1 structure. Reversing mutant mis-localization by appending a NLS increased toxicity of the mutant in a motor neuron-like mammalian cell line and in neurons of *C. elegans*, suggesting that misfolded SOD1 is more toxic in the nucleus than in the cytosol. Evidence is offered to suggest altered nucleocytoplasmic transport when mutant SOD1 is retained in the nucleus. The authors suggest that exposure of NES-like regions in misfolded proteins might be a defense mechanism to reduce their toxicity in the nucleus.

All three referees agreed that the first part of the paper, documenting exposure of a cryptic and buried NES in numerous SOD1 mutants as the basis of altered localization, was thorough, convincing, and represented a new contribution to this field. However, the relevance of this observation for either normal cellular physiology or disease pathogenesis was not sufficiently clear from the experiments thus far, and alternative explanations for the observed changes in toxicity were raised. Furthermore, all the referees felt that the conclusion of altered nucleocytoplasmic transport was premature and insufficiently explored. Finally, one referee felt that the basis of mutant-dependent exposure of the NES remained to be clarified in some greater detail. On the basis of these comments, we recommend the following three areas for improvement.

Essential revisions:

1) A major conclusion of the study is that nuclear SOD1 is more toxic than cytosolic SOD1, suggesting that nuclear export by the mechanism described here is protective. However, the experiments to demonstrate this are not optimal, and some controls are missing. The authors have enforced wild type-like localization of mutant SOD1 by addition of a NLS. The consequence is that this rather abundant and over-expressed protein is constitutively imported and exported endlessly, thereby imposing a greater burden on the nuclear export machinery than would ordinarily be the case in the absence of a NLS. Thus, the increased toxicity might well be due to this (or to the appended NLS somehow having a toxic effect), and not necessarily to SOD1 localization per se. The more suitable and precise experiment is to determine whether L42Q (which is in fact the normal variant in mouse and not expected to disrupt folding by itself) changes the toxicity of G85R or other SOD1 mutants. By doing this, normal wild type-like localization is restored without adding any extra burden to the cell's trafficking pathways. In both the NLS and L42Q experiments, it is crucial to include wild type SOD1 as a control. This would ensure that any effects on toxicity are mutant-specific. These experiments in cultured mammalian cells are straightforward and the authors have all the tools to do it in a reasonable span of time. The referees and editor acknowledge that the analogous experiments in worms would take longer, but it is sufficiently important to wait for such results even if it exceeds the two-month revision timeline.

2) The experiment to suggest that nucleocytoplasmic transport is affected by SOD1 mutants (Figure 6) is incomplete and rather preliminary. The observation of steady state differences in polyA FISH signal is insufficient to conclude an effect on transport, since other explanations such as altered mRNA processing or polyadenylation are possible. If the authors wish to make conclusions about transport, then the authors need to measure transport more directly. Furthermore, a key control in any experiments aimed at elucidating toxicity mechanisms would be to ensure that the effect is due to misfolded SOD1, and not just a highly-expressed NES-containing protein. Thus, it seems important to test whether their mCherry-P1 and mCherry-P2 constructs have any effect on mRNA localization (the prediction being that neither would have any effect in the authors' model, but P1 would have an effect in the alternative model above). The referees acknowledge that properly elucidating the effect on nucleocytoplasmic transport as a potential contributor to toxicity would require extensive studies. Thus, if the authors are not prepared to pursue such investigations for this paper, the recommendation is to remove Figure 6 and only speculate in the Discussion about toxicity mechanisms.

3) An important unanswered question is whether the exposure of NES by SOD1 mutations is an intrinsic property of the mutant protein, or is the result of an unfolding activity from the cell. A key experiment that may distinguish between these possibilities is to determine whether recombinant SOD1 purified from *E. coli* has the NES exposed in a mutant-specific manner, or if this is only a feature observed in the cellular context. Such an experiment would provide greater insight into the mechanism of NES exposure.

---

## [Author Response]

*Essential revisions:*

*1) A major conclusion of the study is that nuclear SOD1 is more toxic than cytosolic SOD1, suggesting that nuclear export by the mechanism described here is protective. However, the experiments to demonstrate this are not optimal, and some controls are missing. The authors have enforced wild type-like localization of mutant SOD1 by addition of a NLS. The consequence is that this rather abundant and over-expressed protein is constitutively imported and exported endlessly, thereby imposing a greater burden on the nuclear export machinery than would ordinarily be the case in the absence of a NLS. Thus, the increased toxicity might well be due to this (or to the appended NLS somehow having a toxic effect), and not necessarily to SOD1 localization per se. The more suitable and precise experiment is to determine whether L42Q (which is in fact the normal variant in mouse and not expected to disrupt folding by itself) changes the toxicity of G85R or other SOD1 mutants. By doing this, normal wild type-like localization is restored without adding any extra burden to the cell's trafficking pathways. In both the NLS and L42Q experiments, it is crucial to include wild type SOD1 as a control. This would ensure that any effects on toxicity are mutant-specific. These experiments in cultured mammalian cells are straightforward and the authors have all the tools to do it in a reasonable span of time. The referees and editor acknowledge that the analogous experiments in worms would take longer, but it is sufficiently important to wait for such results even if it exceeds the two-month revision timeline.*

We agree with the reviewers that enforced wild type-like localization of mutant SOD1 by addition of a NLS could change it into a nuclear-cytoplasmic shuttling protein, which may impose a greater burden on the nuclear export machinery and thus may alter the toxicity of mutant SOD1. Thanks to the reviewers for suggesting the more suitable and precise experiment, that is, to use the L42Q substitution to restore mutant SOD1 nuclear localization instead of amending a NLS. Using this approach, we have confirmed that, as predicted, L42Q substitution restored SOD1^G85R^ nuclear localization (Figure 6). Using an anti-NLP immunoprecipitation approach (under native conditions), we demonstrated that the L42Q substitution does not cause exposure of the NES-like sequence in wild type SOD1 (Figure 6). L42Q substitution also did not change the amount of SOD1^G85R^ immunoprecipitated by anti-NLP, suggesting that it does not alter SOD1^G85R^ structure (Figure 6). We also showed that the toxicity of SOD1^L42Q^ to NSC34 cells is similar to that of wild type SOD1 (Figure 6). These results strongly support that L42Q substitution has minimal effects on SOD1 folding. Therefore, SOD1^G85R/L42Q^ differs from SOD1^G85R^ only in its ability to be exported from the nucleus. In sum, these results set the stage for further testing the toxicity of misfolded nuclear SOD1 by comparing SOD1^G85R^ (misfolded, exported) versus SOD1^G85R/L42Q^ (misfolded, retained) in a *C. elegans* model of ALS.

We generated several new transgenic *C. elegans* lines: SOD1^G85R/L42Q^ and the control SOD1^L42Q^. Three independent lines each for SOD1 variant were made, and these animals were compared with SOD1^G85R^ and wild type SOD1 lines using several approaches, including subcellular localization, effects on locomotion (body bending), and survival. All of the SOD1 proteins were tagged with YFP for direct observation by fluorescence microscopy. We found that SOD1^L42Q^ behaves similarly to wild type SOD1 for all aspects examined, which further confirms the L42Q substitution does not cause misfolding of SOD1 (Figure 7–Figure 9). As observed in cultured cells, SOD1^G85R^ was cleared from the nuclei of motor neurons in both L1 and adult *C. elegans* (Figure 7). In contrast, SOD1^G85R/L42Q^ was localized in both the cytoplasm and the nuclei in motor neurons in L1 *C. elegans* (Figure 7). In adult *C. elegans*, the nuclear localization of SOD1^G85R/L42Q^ was only observed in a fraction of the neurons (Figure 7). A possible explanation is that cytoplasmic aggregation of the mutant protein prevents its entry into the nucleus. Indeed, we found that in adult *C. elegans*, like SOD1^G85R^, SOD1^G85R/L42Q^ formed large cytoplasmic aggregates (Figure 7). Biochemical characterization demonstrated that SOD1^G85R^ and SOD1^G85R/L42Q^ lines express the transgene at similar levels and also form similar amount of insoluble aggregates (Figure 7). These results support that SOD1^G85R/L42Q^ differs from SOD1^G85R^ only in its nuclear localization.

We obtained two lines of evidence suggesting that nuclear localization of SOD1^G85R^ is toxic to neurons in *C. elegans*. First, three lines of SOD1^G85R/L42Q^ worms all exhibited more severe defects in locomotion compared to SOD1^G85R^ animals (Figure 7). As controls, three lines of SOD1^L42Q^*C. elegans* showed body bending rates similar to their wild type counterparts (Figure 7). Second, there was a strong effect on survival for all three SOD1^G85R/L42Q^ lines, where lifespan decreased compared to the SOD1^G85R^ line (Figure 8). As controls, all three SOD1^L42Q^ lines were found to have similar survival curves as the wild type SOD1 line (Figure 8). In search for the potential cause of death, we found that about 70% of SOD1^G85R/L42Q^ animals had egg-laying defects with an average of 4.2 eggs/worm hatched inside mothers, known as “bagging”. The bagging defect for SOD1^G85R/L42Q^ was significantly worse than was found for SOD1^G85R^ animals (Figure 9). About 17% of SOD1^G85R^ animals had an average of 2.4 eggs/worm hatched inside mothers (Figure 9). It is likely that loss of muscle function caused by severe motor neuron degeneration is responsible for these bagging phenotypes in SOD1^G85R/L42Q^ animals. Future studies are warranted to understand the mechanisms underlying the bagging phenotype observed for SOD1^G85R/L42Q^ animals.

These results confirm that restoring nuclear localization of SOD1^G85R^, via an L42Q substitution, can cause additional toxicity, whereas nuclear export of misfolded SOD1 has a protective effect to the cells. Because the potential complex effects associated with the amended NLS, as pointed out by the reviewers, we have removed all data using NLS-tagged SOD1^G85R^ in this revised manuscript.

*2) The experiment to suggest that nucleocytoplasmic transport is affected by SOD1 mutants (Figure 6) is incomplete and rather preliminary. The observation of steady state differences in polyA FISH signal is insufficient to conclude an effect on transport, since other explanations such as altered mRNA processing or polyadenylation are possible. If the authors wish to make conclusions about transport, then the authors need to measure transport more directly. Furthermore, a key control in any experiments aimed at elucidating toxicity mechanisms would be to ensure that the effect is due to misfolded SOD1, and not just a highly-expressed NES-containing protein. Thus, it seems important to test whether their mCherry-P1 and mCherry-P2 constructs have any effect on mRNA localization (the prediction being that neither would have any effect in the authors' model, but P1 would have an effect in the alternative model above). The referees acknowledge that properly elucidating the effect on nucleocytoplasmic transport as a potential contributor to toxicity would require extensive studies. Thus, if the authors are not prepared to pursue such investigations for this paper, the recommendation is to remove Figure 6 and only speculate in the Discussion about toxicity mechanisms.*

We agree with the reviewers that the data to suggest that nucleocytoplasmic transport is affected by SOD1 mutants (Figure 6) was rather preliminary. As suggested by the reviewers, we have removed this data and briefly discussed this possibility in the Discussion section.

*3) An important unanswered question is whether the exposure of NES by SOD1 mutations is an intrinsic property of the mutant protein, or is the result of an unfolding activity from the cell. A key experiment that may distinguish between these possibilities is to determine whether recombinant SOD1 purified from E. coli has the NES exposed in a mutant-specific manner, or if this is only a feature observed in the cellular context. Such an experiment would provide greater insight into the mechanism of NES exposure.*

As discussed with the editor, we have clarified how the experiments were done for Figure 3 and stated that GST-tagged SOD1 mutants, but not wt SOD1, interact with recombinant CRM1.

Disruption of the NES-consensus in SOD1 mutants also abolished their interaction with CRM1 (Figure 3). As suggested by the editor, we have now added new data showing that the interaction between CRM1 and SOD1 mutants is RanGTP-dependent using recombinant proteins (Figure 3). These results favor a model in which the exposure of NES by SOD1 mutations is an intrinsic property of the mutant protein.